# Therapy-Resistant Acute Myeloid Leukemia Stem Cells Are Resensitized to Venetoclax + Azacitidine by Targeting Fatty Acid Desaturases 1 and 2

**DOI:** 10.3390/metabo13040467

**Published:** 2023-03-24

**Authors:** Rachel Culp-Hill, Brett M. Stevens, Courtney L. Jones, Shanshan Pei, Monika Dzieciatkowska, Mohammad Minhajuddin, Craig T. Jordan, Angelo D’Alessandro

**Affiliations:** 1Department of Biochemistry and Molecular Genetics, University of Colorado Denver, Anschutz Medical Campus, Aurora, CO 80045, USA; 2Division of Hematology, University of Colorado Denver, Anschutz Medical Campus, Aurora, CO 80045, USA; 3Department of Medical Biophysics, University of Toronto Princess Margaret Cancer Center, Toronto, ON M5G 1L7, Canada

**Keywords:** acute myeloid leukemia, leukemic stem cells, fatty acid metabolism, lipidomics

## Abstract

Recent advances in targeting leukemic stem cells (LSCs) using venetoclax with azacitidine (ven + aza) has significantly improved outcomes for de novo acute myeloid leukemia (AML) patients. However, patients who relapse after traditional chemotherapy are often venetoclax-resistant and exhibit poor clinical outcomes. We previously described that fatty acid metabolism drives oxidative phosphorylation (OXPHOS) and acts as a mechanism of LSC survival in relapsed/refractory AML. Here, we report that chemotherapy-relapsed primary AML displays aberrant fatty acid and lipid metabolism, as well as increased fatty acid desaturation through the activity of fatty acid desaturases 1 and 2, and that fatty acid desaturases function as a mechanism of recycling NAD+ to drive relapsed LSC survival. When combined with ven + aza, the genetic and pharmacologic inhibition of fatty acid desaturation results in decreased primary AML viability in relapsed AML. This study includes the largest lipidomic profile of LSC-enriched primary AML patient cells to date and indicates that inhibition of fatty acid desaturation is a promising therapeutic target for relapsed AML.

## 1. Introduction

Acute myeloid leukemia (AML) is a cancer derived from the myeloid lineage of blood cells. It is the most common acute leukemia in adults, with nearly 120,000 global cases per year [1]. AML is characterized by the overproduction of leukemic blasts, which prevent the function and development of healthy hematopoietic cells [2]. Despite advances in supportive care and targeted treatments, five-year survival rates remain extremely low. When conventional chemotherapeutics are tolerated, they eliminate most proliferating bulk tumor cells. However, disease-initiating leukemic stem cells (LSCs) are not always eradicated, leading to disease progression and relapse in over 68% of cases [3]. A 10- to 100-fold expansion of the LSC population often exacerbates disease relapse [4], after which the overall five-year survival rate falls to just 10% [5]. Consequently, LSC eradication is crucial to allowing complete remission in relapsed patients and prevent relapse in previously untreated (de novo) patients.

LSCs are a well-established therapeutic target in AML [6], but heterogeneity presents challenges to identifying effective therapeutics. Recent studies suggest that targeting the unique, conserved metabolic profile of LSCs increases the likelihood for successful eradication [7,8]. While most cancer cells are reliant on glycolysis [9], LSCs preferentially use oxidative phosphorylation (OXPHOS) for survival [7]. In de novo LSCs, OXPHOS is driven by amino acid metabolism. Pre-clinical studies in de novo AML demonstrate that inhibition of B-cell lymphoma 2 (BCL-2) decreases amino acid metabolism and inhibits OXPHOS in LSCs, resulting in preferential LSC elimination and underlining the importance of targeting LSC metabolism. The combination of venetoclax, a Bcl-2 homology domain 3 (BH3) mimetic and selective BCL-2 inhibitor, with azacitidine, a hypomethylating agent, shows promising results as a novel clinical strategy to improve AML therapy [10,11].

Although BCL-2 inhibition is effective for de novo AML patients, ~79% of patients who have relapsed from chemotherapy [12] display venetoclax resistance [13]. Relapsed/refractory (R/R) LSCs survive by maintaining functional OXPHOS after treatment with (ven + aza) [7,14], and display increased fatty acid levels, uptake, and metabolism into tricarboxylic acid (TCA) cycle intermediates [13]. Decreased viability upon inhibition of fatty acid oxidation [13] suggests R/R LSCs compensate for a loss of amino acids upon BCL-2 inhibition through the upregulation of fatty acid metabolism, driving OXPHOS and LSC survival [7]. However, while this has never been explored in AML, studies suggest fatty acid desaturation may be crucial to therapeutic resistance in other cancers: increased unsaturated fatty acids have been observed in cancer stem cells, and the inhibition of stearoyl-CoA desaturase-1 (SCD-1) results in decreased proliferation [15]. A novel mechanism of fatty acid desaturation has also been identified that contributes to cancer stem cell plasticity [16], and very long chain fatty acid metabolism is critical to LSC mitochondrial metabolism [17]. Further, recent studies show NAD+ salvage is critical for unsaturated fatty acid synthesis [18] and AML pathogenesis [19], and fatty acid desaturation may act as a mechanism for NAD+ recycling [20]. As a result, targeting fatty acid desaturation may offer a therapeutic strategy for AML and other related cancers. In the present study, we interrogate the role of fatty acids, lipids, and fatty acid desaturation in the mechanism of survival for phenotypically primitive primary AML cells in relapsed AML using primary AML specimens.

## 2. Experimental Design

Extensive details are provided in the Appendix A.

### 2.1. Cell Sorting

Primary AML specimens were thawed, stained with CD45 (571875, BD Biosciences, San Jose, CA, USA) to identify the blast population, CD19 (555413, BD Biosciences) and CD3 (557749, BD Biosciences) to exclude lymphocyte populations, DAPI (278298, EMD Millipore, Rockville, MD, USA), and CellROX deep red (C10422, Thermo Fisher, Waltham, MA USA), and sorted using a BD FACSARIA (BD Biosciences). ROS-Low LSCs were identified as the cells with the 20% lowest ROS, as previously described [21].

### 2.2. Global UHPLC-MS Metabolomics, Lipidomics, and Metabolic Tracing

ROS-Low w LSCs were enriched, as described above. M9-ENL1 cells were prepared as previously described [22]. Analyses were performed via UHPLC-MS, as extensively described in the Appendix A. Metabolite assignment was performed against an in-house standard library, as reported [23]. Metabolic tracing was performed using U13C-glucose and U13C-linoleic acid (Cambridge Isotope Laboratories, Andover, MA, USA).

### 2.3. Viability Assays

Patient samples were cultured with the indicated treatments for 24 h. As indicated, cells were treated with 500 nM venetoclax, 2.5 μM azacitidine (ABT-199), 5 μM CP24,879 (Cayman Chemical, Ann Arbor, MI, USA), and 0.5 μM SC26196 (Cayman Chemical). All treatments were diluted in DMSO such that the final concentration of DMSO ≤ 0.5%. After treatment, cells were washed with ice-cold FACS buffer (PBS with 0.5% FBS) and stained for 15 min at 4 °C in an FACS buffer containing antibodies against DAPI (278298, EMD Millipore) and Annexin V (556419, BD Biosciences). Stained cells were resuspended in a 1 × Annexin V buffer containing DAPI and 0.5% FBS and analyzed on a BD FACSCelesta (BD Biosciences). Viability was determined by the percentage of Annexin V- and DAPI- cells within the parent population. Data were analyzed and prepared via FlowJoTM (FlowJo LLC, Ashland, OR, USA).

### 2.4. Transfection of siRNA in MOLM-13 AML Cells

MOLM-13 cells grown in RPMI media were transfected with siRNA constructs targeting FADS1, FADS2, or a non-targeting scrambled siRNA (Dharmacon, Lafayette, CO, USA), following established protocols [24]. Specifically, 2 × 10^6^ cells were electroporated using the Neon electroporator (Invitrogen, Waltham, MA, USA) in Buffer T: R 1600 V, 10 ms, 3 pulses. The MOLM-13 cells were generously provided by the DeGregori lab (CU Anschutz).

### 2.5. Quantitative RT-PCR

RNA was isolated with the RNeasy plus mini kit (QIAGEN, Germantown, MD, USA). cDNA was synthesized using the iScript One-Step RT-PCR kit (Bio-Rad, Hercules, CA, USA). A quantitative real-time PCR was performed with LightCycler96 real-time PCR, using SYBR Green I Master Mix reagent (Roche Applied Science, Basel, Switzerland).

### 2.6. Isolation and Pulldown of FADS2 in Primary AML

Primary AML specimens were lysed using CytoBuster™ Protein Extraction Reagent (Millipore Sigma, Rockville, MD, USA). DynabeadsTM Protein G Immonuprecipitation Kit was utilized according to manufacturer instructions, using polyclonal Rabbit FADS2 (Invitrogen, Waltham, MA, USA) and Recombinant monoclonal Rabbit IgG (Abcam, Boston, MA, USA).

### 2.7. Proteomics Global Analysis

Samples were analyzed using the timsTOF Pro mass spectrometer (Bruker Daltonics, Bremen, Germany) via a nano-electrospray ion source (Captive Spray, Bruker Daltonics, Bremen, Germany), as described in the Appendix A. Peptide spectral matching was performed against the Uniprot human database.

### 2.8. Quantification and Statistical Analysis

Error bars represent the standard deviation (SD). Biological factors were investigated for significance using a two-way ANOVA and the two-tailed Student’s *t*-test with paired or unpaired analysis where appropriate. Specific analyses are indicated in the figure legends. A *p*-value less than 0.05 was considered significant. Data with statistical significance (* *p* < 0.05, ** *p* < 0.01, *** *p* < 0.001, **** *p* < 0.001) are shown in the Figures. All statistical analyses were performed using the Graph Pad Prism 9 statistical package (GraphPad Software, San Diego, CA, USA) or Metaboanalyst (University of Alberta, Edmonton, AB, Canada).

## 3. Results

### 3.1. Relapsed AML Displays Aberrant Fatty Acid Metabolism

Our previous work demonstrated that R/R AML LSCs display an increased fatty acid metabolism as well as functional OXPHOS upon treatment with ven + aza, suggesting R/R LSCs use fatty acids as a mechanism of maintaining OXPHOS and therefore survival in AML [7]. Further, we more recently reported that resistance to ven + aza occurs as a consequence of an upregulated fatty acid metabolism [13]. Since polyunsaturated fatty acids (PUFAs) are more readily oxidized than saturated fatty acids (SFAs) [25], we hypothesized that the increased fatty acid dependency of relapsed AML LSCs could be explained by changes in free and conjugated fatty-acyl composition in the lipidome. To investigate this hypothesis, we applied metabolomics and lipidomics analyses to ROS-Low primary AML cells obtained from patients and isolated by flow cytometry, as previously described [21]. (Figure 1A) As shown in Figure 1B, we confirmed previous observations of aberrant free fatty acids in relapsed AML cells. Notably, elevated fatty acids were only evident in the LSC-enriched population and not in the blast population. (Appendix A). Most fatty acids were increased in relapsed AML cells; however, increases in several highly unsaturated long-chain fatty acids (HUFAs), or fatty acids containing at least 20 carbons and three unsaturations were of interest [26]. These include arachidonic acid (20:4, *p* < 0.0001) and eicosapentaenoic acid (20:5, *p* < 0.01), both of which were consistently increased in all relapse samples. (Figure 1B) Interestingly, these HUFAs are products of the ω3 and ω6 fatty acid desaturation pathways, driven by fatty acid desaturase 1 (FADS1) and 2 (FADS2) [27]. Levels of these and other HUFAs within the ω3 and ω6 desaturation pathways are shown in Figure 1C. Significantly increased linoleic acid (18:2, *p* < 0.01), dihomo-γ-linoleic acid (20:3, *p* < 0.01), and docosapentaenoic acid (22:5, *p* < 0.001), as well as those previously mentioned, suggest increased activity of both FADS1 and FADS2 in relapsed AML. These data indicate that relapsed ROS-Low AML cells may have increased activity of the ω3 and ω6 fatty acid desaturation pathways, driven by FADS1 and FADS2. 

### 3.2. Relapsed AML Also Displays Aberrant Lipid Metabolism

Considering the increase in unsaturated free fatty acids, we set out to determine whether this phenomenon also impacted conjugated fatty acids in the lipidome. Aberrant lipid metabolism has been well documented in cancer stem cells (CSCs) as a mechanism to maintain stemness and fulfill biomass and energetic demands [28]. To better understand the extent of these aberrations in relapsed AML, we performed a lipidomics screen in de novo and relapsed ROS-Low AML cells using LipidSearchTM. To the best of the authors’ knowledge, this is the first comprehensive lipidomics profile performed in LSC-enriched primary AML patient cells. Lipids were categorized into classes based on lipid head composition (Table 1).

Almost all identified lipid classes were significantly increased in relapsed AML, including sphingomyelins (SM), phosphatidylinositols (PI), and triglycerides (TG). Concurrently, others were significantly decreased or unchanged in relapsed primary AML cells, including lysophosphatidylcholines (LPC), lysophosphatidylethanolamines (LPE), and (O-Acyl)-ω-hydroxy fatty acids (OAHFA). These data suggest aberrations in the lipid metabolism may be relevant to the relapsed AML phenotype. (Figure 2A) As shown in Figure 2B, most significantly altered lipids occurred in relapsed AML, and notably, many contain HUFAs. To determine whether unsaturated fatty acids impacted the most significantly altered lipids in relapsed AML, we enriched the top 20 most significantly affected lipids increased in relapsed AML but not de novo. Notably, 15 of the 20 affected lipids contained at least one conjugated-fatty acyl chain with >2 desaturations. (Figure 2C) These findings suggest fatty acid desaturation may be relevant to relapsed AML pathogenesis. A full list of detected lipid metabolites is provided in the Appendix A.

### 3.3. Relapsed AML Displays Increased Fatty Acid Desaturation

Increased levels of HUFAs have been observed in the plasma of AML patients, especially those with high levels of blasts in the bone marrow or peripheral blood and an unfavorable prognostic risk [29]. Fatty acid desaturation has also been implicated in CSC pathogenesis: an alternative fatty acid desaturation pathway increases their metabolic plasticity and contributes to therapeutic resistance [16]. Further, very long chain fatty acid metabolism was recently shown to be required for leukemia cell mitochondrial metabolism [17]. Therefore, to probe the importance of long chain fatty acid desaturation in the context of relapsed AML, we analyzed the ratio of unsaturated to saturated fatty acids in primary samples obtained from the same patient at diagnosis (de novo) and at relapse. While the ratio was decreased for fatty acids containing 18 carbons, for those containing 20 and 22 carbons, the ratio was significantly increased in relapsed ROS-Low primary AML cells. (Figure 3A) This suggests fatty acid desaturation, especially in long chain fatty acids, may contribute to relapsed AML pathogenesis.

To investigate whether lipids containing unsaturated fatty acids significantly contribute to the aberrant lipid phenotype observed in relapsed ROS-Low primary AML cells, we categorized individual lipids into those containing only saturated fatty acids and those containing only unsaturated fatty acids. We then plotted the two categories separately from the total lipid class to determine whether the unsaturated fatty acids within those classes were significantly altered and whether those lipids containing unsaturated fatty acids contributed to the overall significance of the lipid class, if applicable. As shown in Figure 3B,C, when lipid classes were filtered to display lipids containing fully saturated fatty acids, only five were significantly increased in relapsed AML. However, when classes were filtered to display lipids containing only unsaturated fatty acids, nine were significantly increased in relapse. Together, these data suggest the aberrant lipid phenotype observed in relapsed ROS-Low primary AML cells is at least partly explained by lipids containing only unsaturated fatty acids.

### 3.4. Fatty Acid Desaturase Expression Is Increased in Relapsed AML

To further expand upon the elaboration of data from the Cancer Genome Atlas (TCGA), we performed a single cell RNAseq data analysis on a paired de novo and relapsed AML patient sample. In order to transcriptionally identify the subset of LSCs in these specimens, we employed a previously characterized series of stem cell markers described as the LSC17 score [30] and applied them to paired de novo and relapse patient samples. Consistent with our hypothesis of a role for FADS1 and 2 in LSC metabolic reprogramming, FADS1 and FADS2 expression overlapped with expression patterns of LSC-defining genetic markers (Figure 4B), suggesting FADS1 and FADS2 are overexpressed in self-renewing AML cells and are essential to LSC survival and pathogenesis. To further expand on these data, we then performed a quantitative analysis of FADS1 and FADS2 via immunoprecipitation and a targeted quantitation via nanoUHPLC-MS/MS. While FADS1 was below the limit of detection, we confirmed a 63% increase in the levels of FADS2 in relapsed cells (Appendix A).

### 3.5. Inhibition of Fatty Acid Desaturases 1 and 2 Expression or Activity Sensitizes Relapsed AML to ven + aza

To determine whether FADS1 and FADS2 contribute to ven + aza resistance, we performed the siRNA knockdown of FADS1 and FADS2 with ven + aza in MOLM-13 AML cells and measured their viability. The FADS knockdown was performed successfully (Appendix A). As is shown in Figure 5A, ven + aza treatment with SCR resulted in a 17% decrease in viability. However, when combined with FADS1 or FADS2 knockdown, the addition of ven + aza resulted in a significant 29% and 33% decrease in viability, respectively. The decreased viability with ven + aza suggests that FADS1 and FADS2 play a role in AML cell survival. To determine whether FADS1 and FADS2 contribute to ven + aza resistance in relapsed AML, we observed viability in de novo and relapsed ROS-Low primary AML cells using pharmacological FADS1 and FADS2 inhibitors. CP24,879 targets both FADS1 and 2, while SC26196 is a selective FADS2 inhibitor. To the best of our knowledge, there are no commercially available specific inhibitors of FADS1 alone. We first set out to determine whether the FADS1/2 and FADS2 inhibitors successfully targeted FADS. To this end, we performed metabolic tracing using stable heavy isotope labeled 13C18-linoleic acid (18:2n6), the substrate for FADS2. The FADS2 product from this substrate, 13C18-γ-linoleic acid (18:3n6), was significantly decreased upon treatment with both inhibitors, suggesting they successfully inhibit FADS2 activity. (Figure 5B). We then incubated ROS-Low primary AML cells with the inhibitors and ven + aza. When combined with ven + aza, treatment with the FADS1/2 inhibitor significantly decreased viability in both de novo and relapsed ROS-Low primary AML cells, while FADS2 inhibition significantly decreased viability only in de novo cells. (Figure 5C) This recapitulates the outcome of genetic FADS1 and FADS2 inhibition in MOLM-13 AML cells and suggests that FADS1 and/or FADS2 may offer a novel pharmacologic target for AML.

### 3.6. FADS1, FADS2 Inhibition Decreases Metabolic Flux into the TCA Cycle

Previous studies show that fatty acids compensate for a decrease in amino acids upon treatment with ven + aza [7,13], providing fuel for the TCA cycle and OXPHOS. We therefore hypothesized that FADS1 and FADS2 contribute to AML survival through the production of HUFAs, which fuel the TCA cycle and OXPHOS. We therefore performed metabolic tracing in AML cells [31] using 13C18-linoleic acid with FADS1/2 or FADS2 pharmacologic inhibitors and observed elongation and desaturation through the ω6 pathway as well as incorporation into the TCA cycle. As shown in Figure 6, we observed significantly decreased levels of all FADS1 and FADS2 products within the ω6 pathway upon their inhibition. Additionally, accumulation of 13C18-eicosadienoic acid, which results from the elongation of 13C18-linoleic acid, was observed upon FADS inhibition. Notably, 13C2-citrate decreased upon FADS inhibition, the production of which results from oxidation of 13C18-linoleic acid or its downstream products. However, based on our hypothesis that ω6 pathway products fuel the TCA cycle, we expected a more dramatic reduction of labeled citrate. These results instead suggest that HUFAs are not the primary TCA cycle fuel source and instead may be affecting TCA cycle metabolism through an alternative mechanism.

To further investigate these findings, we performed additional metabolic tracing in AML cells using 13C18-linoleic acid upon FADS1 or FADS2 siRNA knockdown. (Appendix A) We recapitulated the decrease in ɣ-linolenic acid (18:3), the accumulation of eicosadienoic acid (20:2), and an increase in the ratio of eicosadienoic acid (20:2) to linoleic acid (18:2), suggesting increased elongase activity. Surprisingly, however, AML cells exhibited increased TCA cycle activity upon the genetic inhibition of FADS1 or FADS2. Here, genetic inhibition may prevent linoleic acid desaturation to such an extent that it increases its availability for use in the TCA cycle. Taken together, these data suggest the mechanism by which FADS1 and FADS2 contribute to AML pathogenesis is not through the production of HUFAs directly.

### 3.7. FADS1, FADS2 Inhibition Decreases Glycolytic Activity

As FADS1 and FADS2 inhibition did not result in decreased HUFA incorporation into the TCA cycle as expected, we performed a global metabolomics analysis in primary relapsed patient AML cells to determine the metabolic effects of FADS inhibition. As shown in Figure 7A,B, FADS inhibition resulted in significant metabolic alterations. We observed an expected decrease in several fatty acids, but additional alterations were observed in glycolytic and TCA cycle intermediates (Appendix A), including increased glyceraldehyde 3-phosphate and succinate and increased ratios of succinate to fumarate and citrate to 2-oxoglutarate. As a result, we performed metabolic tracing in ROS-Low primary AML cells using U13C-glucose (Figure 7C). We observed significantly increased fructose 1,6-bisphosphate, decreased 1,3-bisphosphoglycerate, and a significantly increased ratio of fructose 1,6-bisphosphate to 1,3-bisphosphoglycerate upon FADS inhibition, suggesting increased glycolytic activity, specifically of GAPDH. We also observed significantly decreased levels of the TCA cycle intermediate citrate upon FADS inhibition and, while not significant, decreased succinate, increased fumarate, and a decreased succinate to fumarate ratio. The conversion of succinate to fumarate is driven by succinate dehydrogenase (SDH); therefore, these data suggest decreased SDH activity upon FADS inhibition. Together, these data show that the inhibition of FADS in relapsed AML decreases glycolytic and TCA cycle activity, especially metabolic enzymes such as GAPDH and SDH.

## 4. Discussion

Venetoclax with azacitidine has dramatically improved the standard of care for older, newly diagnosed (de novo) AML patients [11]. However, R/R AML patients display venetoclax resistance and therefore have limited treatment options. Our previous work identified fatty acid metabolism as a mechanism by which relapsed LSCs exhibit venetoclax resistance [13]. Therefore, we performed global metabolomics to compare the fatty acid metabolome of de novo and relapsed ROS-Low primary AML cells. Relapsed AML indeed displays an aberrant fatty acid metabolism: while most fatty acids are increased, HUFAs such as arachidonic acid (20:4) and eicosapentaenoic acid (20:5) are of specific interest (Figure 1B,C). Increased HUFAs have been observed in the plasma of AML patients with an unfavorable prognostic risk [27], indicating that they may play a role in therapeutic resistance and disease severity. HUFA production is driven by FADS1 and FADS2, the rate-limiting enzymes in the ω3 and ω6 fatty acid desaturation pathways. While these enzymes are relatively understudied, they have been implicated in other cancers [32]. Our previous studies also suggested that relapsed AML LSCs use fatty acids as an alternative mechanism to fuel OXPHOS [7], and unsaturated fatty acids have been implicated as a metabolic marker for cancer stem cells [15]. Therefore, we suspected this phenotype may have a similar role in AML pathogenesis.

Fatty acids can be incorporated into higher lipids, and aberrant lipid metabolism has been well documented in CSCs as a mechanism to maintain stemness and fulfill energetic demands [26]. To determine whether relapsed AML displays an aberrant lipid metabolism, we performed a lipidomics screen of de novo and relapsed ROS-Low primary AML cells. Several lipid classes were significantly increased in relapsed AML cells, including phosphatidylinositol (PI), triglyceride (TG), and phosphatidylcholine (PC) (Figure 2A). In addition to phospholipid metabolism being implicated in the regulation of AML growth and stemness [33], PCs are of particular interest: increased levels of cell-membranous PCs containing PUFAs induced p53 phosphorylation through ataxia telangiectasia (ATR) activation [34], implying that PUFAs, including HUFAs, may play important roles in the signaling pathways relevant to cancer metabolism. In addition to our data, which demonstrate that 15 of the top 20 significantly altered lipids contain at least one polyunsaturated fatty acid (Figure 2C) and polyunsaturated fatty acid-containing lipids significantly contribute to the observed lipid class differences in relapsed AML cells (Figure 3C), this suggests HUFAs are highly relevant to the relapsed AML phenotype. Further, to the best of the authors’ knowledge, this is the first observation of the lipidome of LSC-enriched AML primary patient cells.

As discussed, FADS1 and FADS2 are the rate-limiting enzymes in the ω3 and ω6 desaturation pathways that produce HUFAs. As fatty acid aberrations were most significant for the end products of these pathways, we hypothesized that FADS1 and FADS2 may contribute to AML pathogenesis. High FADS1 expression in AML patients significantly correlates with poor clinical outcome. Although not significant, a similar trend is observed for FADS2 expression. (Figure 4A) Additionally, FADS1 and FADS2 are specifically expressed in the stem and progenitor population and display increased RNA and protein expression in relapsed primary patient cells. (Figure 4B, Appendix A) These data suggest FADS1 and FADS2 are specifically contributing to the pathogenesis of relapsed AML LSCs.

To determine whether these data point to the role of FADS in ven + aza resistance, we performed the siRNA knockdown of FADS1 and FADS2 in MOLM-13 cells. (Figure 5A) The FADS1 and FADS2 knockdown in combination with ven + aza treatment resulted in a further decrease in viability. Additionally, treatment with FADS1/FADS2 and FADS2 pharmacologic inhibitors in combination with ven + aza also decreased viability in relapsed ROS-Low primary AML cells. Increased FADS2 expression has been observed in certain cancers and may offer a mechanism of increased metabolic plasticity [16], suggesting increased FADS1 and/or FADS2 activity may result in therapeutic resistance in relapsed AML.

As we previously determined that R/R AML LSCs use fatty acids to fuel the TCA cycle to compensate for a loss of amino acids upon venetoclax treatment, we hypothesized that HUFAs are also used in this manner. Through 13C18-linoleic acid metabolic tracing (Figure 6), we observed significantly reduced 13C labeling of the TCA cycle intermediate citrate upon FADS1 and FADS2 inhibition. However, based on our hypothesis that the end products of the FADS1 and FADS2 pathways are used to fuel the TCA cycle, it was surprising to us that while we observed dramatic differences in HUFAs such as 13C18-arachidonic acid and 13C18-adrenic acid, these differences were not as notable for 13C2-citrate. 13C2-citrate is produced through the oxidation of 13C18-linoleic acid or its downstream products; therefore, if the TCA cycle was primarily being fueled by the end products of the FADS1 and FADS2 pathways we would expect a more remarkable reduction in TCA cycle incorporation. Taken with the surprising increase in TCA cycle incorporation upon genetic knockdown of FADS1 and FADS2 (Appendix A), these data suggest while FADS1 and FADS2 are relevant to R/R AML pathogenesis, the production of HUFAs for oxidative metabolism is not their direct mechanism.

To gain further understanding of the impact of FADS1 and FADS2 on R/R AML metabolism, we performed 13C6-glucose metabolic tracing and observed alterations in glycolytic and TCA cycle metabolism upon pharmacologic FADS1 and FADS2 inhibition. We again observed significantly decreased 13C2-citrate upon FADS inhibition, confirming that FADS activity impacts TCA cycle metabolism. Based on these results, FADS may specifically impact metabolic enzymes such as GAPDH (as shown by the significantly increased ratio of fructose 1,6-bisphosphate to 1,3-bisphosphoglycerate) and SDH (as shown by the decreased ratio of succinate to fumarate), both of which require NAD+ as a cofactor. Previous studies show fatty acid desaturases function as a mechanism to recycle NAD+ [20], and nicotinamide metabolism is critical to R/R AML [19]. These data therefore suggest that FADS contribute to R/R AML pathogenesis through NAD+ production, which is used to drive metabolic pathways such as glycolysis and the TCA cycle.

## 5. Conclusions

Together, our findings suggest increased FADS activity in relapsed AML LSCs results in the increased production of HUFAs; however, HUFAs do not directly fuel the TCA cycle through oxidation. Instead, FADS1 and FADS2 may function as a mechanism of NAD+ recycling which allows for the continued function of NAD+-dependent metabolic enzymes, such as GAPDH and SDH. Further, this metabolic phenotype can be abrogated by the pharmacologic inhibition of FADS1 and FADS2, with a concomitant effect on viability. Therefore, fatty acid desaturase inhibition may offer a novel therapeutic approach for targeting therapeutically resistant LSCs from AML patients who have relapsed from chemotherapy.

## Figures and Tables

**Figure 1 metabolites-13-00467-f001:**
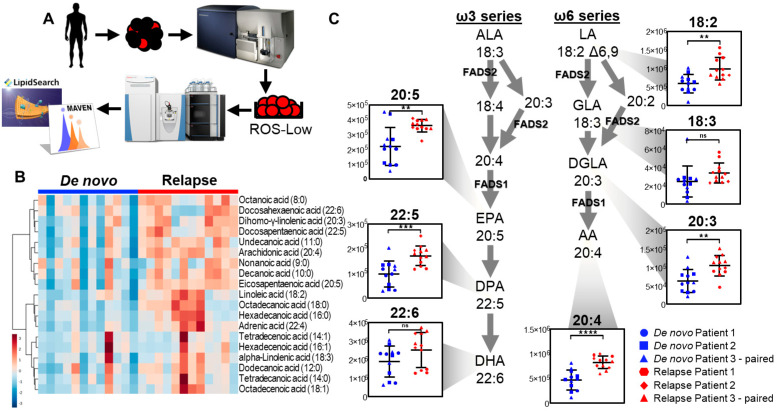
Relapsed LSCs display an aberrant fatty acid metabolism. (**A**) De novo and relapse AML cells were obtained from AML patient leukapheresis product. Flow cytometry was used to isolate ROS-Low leukemic stem cells (LSCs). LSCs were then subjected to mass spectrometry for global metabolomics and lipidomics analyses. (**B**) Heatmap of free fatty acid levels in de novo and relapsed AML LSCs. Metabolomics analysis was performed with n = 3 patient samples with 4 technical replicates. (de novo patients 1, 2, and 3 and relapsed patients 1, 2, and 3). (**C**) Free fatty acids within the ω3 and ω6 fatty acid desaturation pathways in de novo and relapsed AML LSCs. Metabolomics analysis was performed with n = 3 patient samples with 4 replicates. Patient samples are identified using different shapes, with the single paired patient sample both identified as triangles (de novo patients 1, 2, and 3 and relapsed patients 1, 2, and 3). ns *p* > 0.05, ** *p* ≤ 0.01, *** *p* ≤ 0.001, **** *p* ≤ 0.0001.

**Figure 2 metabolites-13-00467-f002:**
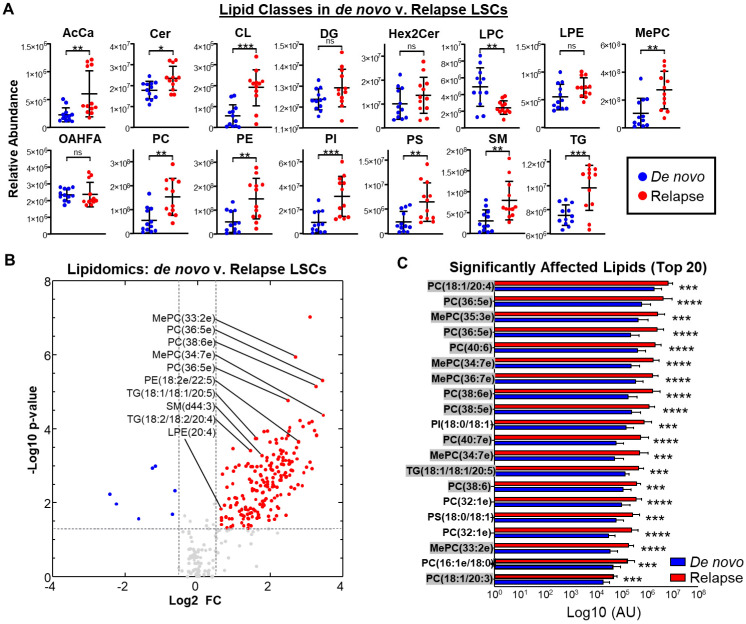
Relapsed LSCs display an aberrant lipid metabolism. Lipidomics were performed using LSCs from de novo patients 1, 2, and 3 and relapsed patients 1, 2, and 3. (**A**) Total levels of lipids within individual lipid classes between de novo and relapsed AML LSCs. (**B**) Volcano plot displaying lipids increased in de novo and relapsed AML LSCs. (**C**) Top 20 significantly affected lipids in de novo and relapsed AML LSCs. Those containing at least 2 unsaturations are highlighted in gray. * *p* ≤ 0.05, ** *p* ≤ 0.01, *** *p* ≤ 0.001, **** *p* ≤ 0.0001.

**Figure 3 metabolites-13-00467-f003:**
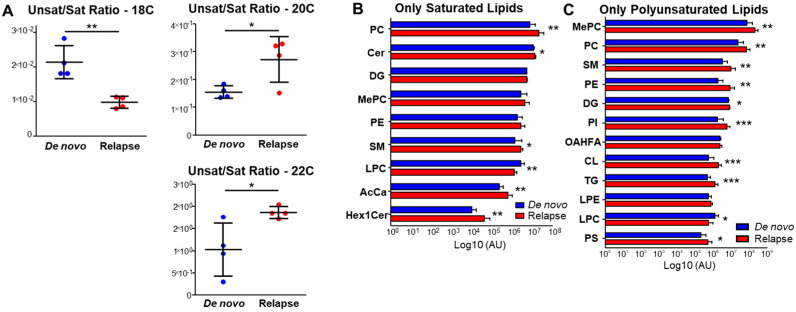
Relapsed LSCs display aberrant fatty acid desaturation. (**A**) Ratio of free fatty acids containing one or more desaturations to fully saturated free fatty acids, for free fatty acids containing 18, 20, and 22 carbons, in paired de novo and relapse patient 3. (**B**) Total levels of lipids containing only saturated fatty acids in de novo and relapsed AML LSCs. (de novo patients 1, 2, and 3 and relapsed patients 1, 2, and 3). (**C**) Total levels of lipids containing only unsaturated fatty acids in de novo and relapsed AML LSCs. (de novo patients 1, 2, and 3 and relapsed patients 1, 2, and 3). * *p* ≤ 0.05, ** *p* ≤ 0.01, *** *p* ≤ 0.001.

**Figure 4 metabolites-13-00467-f004:**
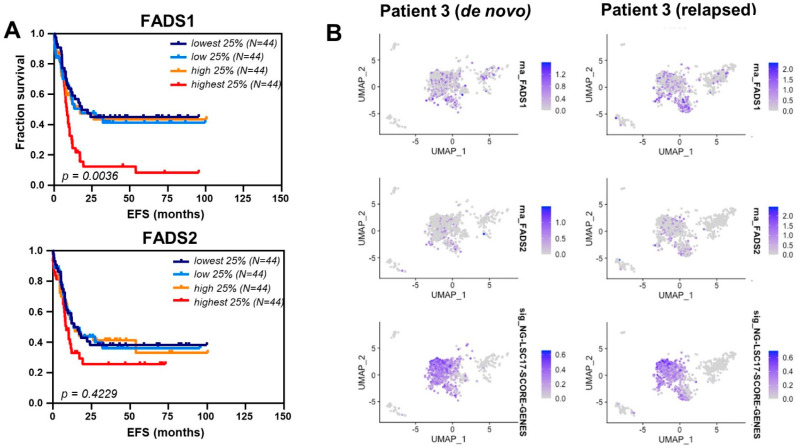
FADS1 and 2 contribute to AML pathogenesis and are expressed in the LSC compartment. (**A**) Fraction of survival of AML patients divided into 4 quartiles of FADS1 and FADS2 expression. Data mined from TCGA. (**B**) Single-cell RNAseq from the paired patient 3 de novo and relapsed samples. Top: Expression of FADS1. Middle: Expression of FADS2. Bottom: cells previously characterized in Ng, S. W., et al. (2016, *Nature*), showing expression of a series of 17 stem cell markers (LSC17).

**Figure 5 metabolites-13-00467-f005:**
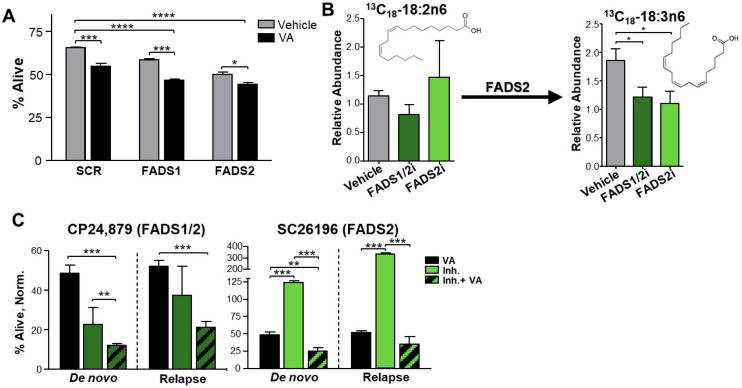
FADS1 and 2 can be genetically and pharmacologically targeted. (**A**) Percentage of MOLM-13 AML cells alive after treatment with scramble siRNA + vehicle; scramble siRNA + 500 nM venetoclax + 2.5 μM azacitidine; FADS1 or FADS2 siRNA + vehicle; and FADS 1 or FADS2 siRNA + 500 nM venetoclax + 2.5 μM azacitidine. (**B**) Relative abundance of 13C18- heavy stable isotope labeled linoleic acid and 13C18- heavy stable isotope labeled gamma-linoleic acid, substrate and product of FADS2, upon treatment with 0.5 μM CP24,879 (FADS1 and FADS2 inhibitor) and 5 μM SC26196 (FADS2 inhibitor) for 8h in primary relapsed patient LSCs (relapse patient 3). (**C**) Percentage of primary de novo and relapsed patient LSCs alive, normalized to vehicle treatment with 500 nM venetoclax + 2.5 μM azacitidine; 500 nM venetoclax + 2.5 μM azacitidine with 0.5 μM FADS1 and 2 inhibitor (CP24,879); and 500 nM venetoclax + 2.5 μM azacitidine with 5 μM FADS2 inhibitor (SC26196) for 24 h (paired patient 3, de novo and relapsed). * *p* ≤ 0.05, ** *p* ≤ 0.01, *** *p* ≤ 0.001, **** *p* ≤ 0.0001.

**Figure 6 metabolites-13-00467-f006:**
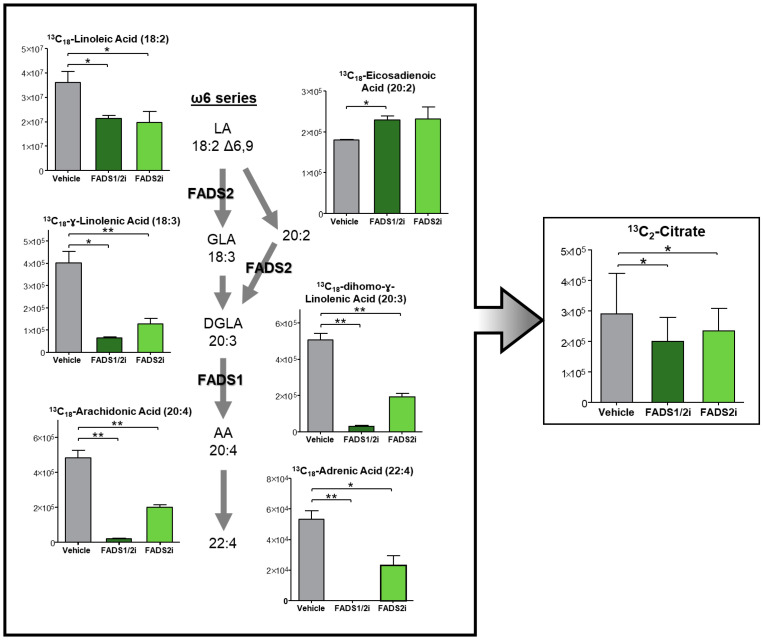
Inhibition of FADS1 and FADS2 decreases products of ω6 pathway and flux into the TCA cycle. Relative abundance after metabolic flux using stable heavy isotope labeled (13C-18) linoleic acid (18:2) at 12 h post treatment with 5 μM CP24,879 (FADS1/2 inhibitor) and 0.5 μM SC26196 (FADS2 inhibitor) in M9-ENL1 cells. * *p* ≤ 0.05, ** *p* ≤ 0.01.

**Figure 7 metabolites-13-00467-f007:**
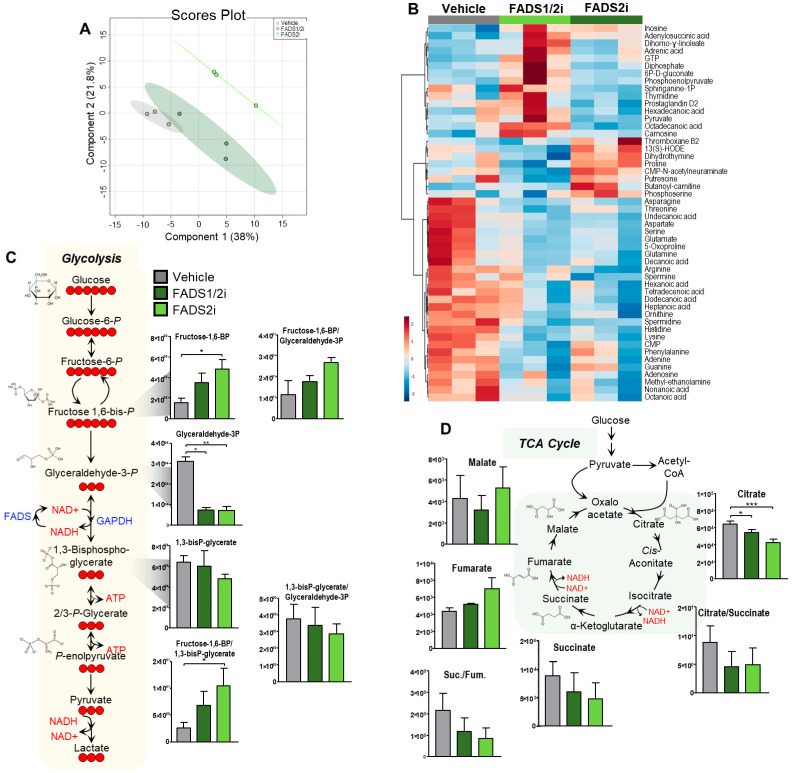
Inhibition of FADS1 and FADS2 leads to alterations in glycolytic metabolism. (**A**) PLSDA comparing bulk AML cells from relapsed patients treated with vehicle, 5 μM CP24,879 (FADS1/2i), or 0.5 μM SC26196 (FADS2i). (**B**) Global heatmap showing the top 50 most significant metabolites by ANOVA, comparing bulk AML cells from relapsed patients treated with vehicle, 5 μM CP24,879 (FADS1/2i), and 0.5 μM SC26196 (FADS2i). (**C**) Levels of U13C glycolytic intermediates after 4 h of incubation with U13C heavy stable isotope labeled glucose in relapsed LSCs with vehicle, 5 μM CP24,879 (FADS1/2i), or 0.5 μM SC26196 (FADS2i). (**D**) Levels of 13C TCA cycle intermediates after 4 h of incubation with U13C heavy stable isotope labeled glucose in relapsed LSCs with vehicle, 5 μM CP24,879 (FADS1/2i), or 0.5 μM SC26196 (FADS2i). * *p* ≤ 0.05, ** *p* ≤ 0.01, *** *p* ≤ 0.001.

**Table 1 metabolites-13-00467-t001:** Lipid class abbreviations.

Lipid Class Abbreviation	Lipid Class Name
AcCa	Acylcarnitine
Cer	Ceramide
CL	Cardiolipin
DG	Diglyceride
Hex2Cer	Hexosylceramide
LPC	Lysophosphatidylcholine
LPE	Lysophosphatidylethanolamine
MePC	Methylphosphocholine
OAHFA	(O-Acyl)-ω-hydroxy fatty acids
PC	Phosphatidylcholine
PE	Phosphatidylethanolamine
PI	Phosphatidylinositol
PS	Phosphatidylserine
SM	Sphingomyelin
TG	Triglyceride

## Data Availability

The data presented in this study are available in the Appendix A for this article.

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
