# Peer review of "Therapy-Resistant Acute Myeloid Leukemia Stem Cells Are Resensitized to Venetoclax + Azacitidine by Targeting Fatty Acid Desaturases 1 and 2"

_metabolites, 2023, doi:10.3390/metabo13040467_

Round 1

Reviewer 1 Report

The work entitled „Therapy resistant acute myeloid leukemia stem cells are resensitized to venetoclax + azacitidine by targeting fatty acid desaturases 1 and 2” is relatively well written and well prepared with adequate methods used herein. However the vital data is missing in some parts and there are some sentences that need to be rephrased and corrected:

I suggest it would be beneficial for the readers to shortly explain sorting strategy for Figure 1 as it is vital for further parts of the manuscript.

Line 88: “Annexin” which one? Why have you used annexin to study cell viability? How the results were obtained and analyzed?

Line 94: „2x106 cells were”

Line 136: „samples. (Fig. 1B) Interest”

Line 208: „those with high bone marrow or peripheral blasts”

Line 214: “in a paired de novo and relapsed patient sample” how these samples were “paired”?

Line 242: “TCGA”? The Cancer Genome Atlas?

Lines 241 – 248 – this part is hard to follow since it is not clear where the data are coming from. Are these new experiments or analysis of some external databases?

Methods on AML cells viability testing are missing: ven concentration? Aza concentration? Timing? pharmacological FADS1 and FADS2 inhibitors? Manufacturer? Dosing? Vehicle?

In figure 5A, why the viability of SCR treated Molm13 cells is so low - approx. 60%?

Do the results in Fig 5C suggest that inhibitor (SC26196) increases cell viability up to 300%?

What is the outcome of FADS pharmacological inhibition in Molm13 cells? How does it correspond to siRNA?

Line 328: what are M9-ENL1 cells?

Author Response

Thank you very much for your comments. Please see below for responses to each point.

I suggest it would be beneficial for the readers to shortly explain sorting strategy for Figure 1 as it is vital for further parts of the manuscript. We have described the sorting strategy in the methods section.

Line 88: “Annexin” which one? Why have you used annexin to study cell viability? How the results were obtained and analyzed? Additional information has been added to the cell viability section of the methods.

Line 94: „2x106 cells were” This error has been corrected to 2x10^6.

Line 136: „samples. (Fig. 1B) Interest” We are unsure what correction is needed here.

Line 208: „those with high bone marrow or peripheral blasts” This sentence has been reworded to improve clarity.

Line 214: “in a paired de novo and relapsed patient sample” how these samples were “paired”? This sentence has been reworded to improve clarity.

Line 242: “TCGA”? The Cancer Genome Atlas? Yes, the full name has been added.

Lines 241 – 248 – this part is hard to follow since it is not clear where the data are coming from. Are these new experiments or analysis of some external databases? This section has been reworded to improve clarity – this is a new experiment. We performed RNAseq on a paired patient sample (de novo and relapse) and applied the previously described LSC17 score to this new sample in addition to identifying FADS.

Methods on AML cells viability testing are missing: ven concentration? Aza concentration? Timing? pharmacological FADS1 and FADS2 inhibitors? Manufacturer? Dosing? Vehicle? Thank you, this information has been added to the figure legends and methods section.

In figure 5A, why the viability of SCR treated Molm13 cells is so low - approx. 60%? Molm13 cells are sensitive to siRNA experiments even using SCR siRNA. Experiments in primary cells/other cell lines were unsuccessful.

Do the results in Fig 5C suggest that inhibitor (SC26196) increases cell viability up to 300%? Yes – this result was replicated and may be a compensatory mechanism, but further experimentation was outside the scope of this manuscript.

What is the outcome of FADS pharmacological inhibition in Molm13 cells? How does it correspond to siRNA? Pharmacological FADS inhibition was not performed in Molm13 cells due to successful experiments in primary AML patient cells.

Line 328: what are M9-ENL1 cells? Information about this cell line has been added to methods.

Reviewer 2 Report

The authors presented a very interesting and well-developed scientific work with a good soundness in the field of metabolomics for cancer research. The AML cohort was small but I think enough for a pilot study as this one. I wish the authors’ will to continue this study in a wider records of AML patients to confirm these preliminary results. For the attractiveness of argument and the significance in the implementation of novel therapeutic approach, in my opinion the paper is suitable for the publication on Metabolites in this form. I’d like just to advise minor stylistic revisions:

-        In the references list, I suggest authors to indicate three authors at least, where many authors are listed, in order to align all the cited papers

-        Check the references 18 and 19 because I found some inaccuracies

-        I did not find the reference 28 in the main text of the paper, please check this one

-        Please check the sentence line 129-131 because I think that something was accidentally left out.

Author Response

Thank you very much for your comments. Please see below for responses to each point.

In the references list, I suggest authors to indicate three authors at least, where many authors are listed, in order to align all the cited papers. The references section has been updated as suggested.

Check the references 18 and 19 because I found some inaccuracies. We are unsure what inaccuracies are described here – these references are correct to the best of our knowledge.

I did not find the reference 28 in the main text of the paper, please check this one. This absence has been corrected.

Please check the sentence line 129-131 because I think that something was accidentally left out. This missed word has been added.

Reviewer 3 Report

In Therapy resistant acute myeloid leukemia stem cells are resen- sitized to venetoclax + azacitidine by targeting fatty acid de-saturases 1 and 2, Rachel Culp-Hill et al. reported that chemotherapy-relapsed primary AML displays aberrant fatty acid and lipid metabolism, as well as increased fatty acid desaturation through the activity of fatty acid desaturases 1 and 2, and fatty acid desaturases function as a mechanism of recycling NAD+ to drive relapsed leukemic stem cells (LSCs) survival. And, both genetic and pharmacologic inhibition of fatty acid desaturation result in decreased primary AML viability in relapsed AML when combined with ven+aza. However, there are several points:

  1. The authors would include the details of cell sorting, including the controls, specifically in the supplementary material.
  2. The culture conditions of Molm-13 cells are not indicated. Moreover, the rationale for its use needs to be clarified.
  3. The inclusion, exclusion, and elimination criteria are not included. What were the criteria to decide if the tumor is relapsed/refractory?
  4. How do differentiate in the samples if HUFAS (arachidonic and eicosapentaenoic acid) are from intracellular metabolism or internalized by tumor cells?
  5. It is recommendable that figure 4 includes the controls of FADS1 and FADS2 expression.
  6. It is recommendable to describe the criteria to establish the FADS1 and FADS2 lowest, low, high, and highest expressions.
  7. Figure 4A was not described in the manuscript.
  8. It is recommendable to include the knockdown controls in figure 5.
  9. In Lane 329: Surprisingly, however
